

# Developing and evaluating a Bayesian weather generator for UK precipitation conditioned on discrete storm types

Paul Bell[1], Jennifer Catto[1], Anne Jones[2], and Stefan Siegert[1]

[1]University of Exeter, Exeter, United Kingdom
[2]IBM Research Europe, The Hartree Centre STFC Laboratory, Sci-Tech Daresbury, Warrington, WA4 4AD, United Kingdom

**Correspondence:** Paul Bell (pb600@exeter.ac.uk)

**Abstract.** Weather generators (WGs) are important tools for downscaling General Circulation Model (GCM) output for climate impact modelling. This study introduces a precipitation WG conditioned on a recent storm types dataset and outlines a methodology for evaluating WGs using proper scoring rules. The storm types are a set of discrete weather types that use atmospheric variables to categorise ERA5 grid cells into fronts, cyclones and thunderstorms or combinations of these. The WG is a Bayesian Generalised Linear Model (GLM) with vertical velocity and humidity as covariates, conditioned on the storm types, and trained on 6 hourly precipitation accumulations. This approach contrasts with previous WGs based on weather types, which use clustering methods such as k-means to generate weather types. A Bayesian model framework is used, instead of typical maximum likelihood approaches, and an informed prior choice is made on observable quantities using the prior predictive distribution. The WG is assessed using proper scoring rules and Diebold-Mariano (DM) significance tests. The use of a DM test to assess the statistical significance of average proper score differences is a key addition to typical WG evaluation approaches, as it helps model developers avoid changes that improve an average score by chance. Calibration is assessed using the probability integral transform histogram and by comparing draws from the posterior predictive distribution to observations. Compared to the same WG not conditioned on storm types the inclusion of storm types improved the average Continuous Ranked Probably Score (CRPS) by a statistically significant amount across the stations considered. When storm types are used as an alternative to continuous atmospheric variables, they provide 33% of the improvement in average CRPS that the atmospheric variables do, averaged over the stations. To quantify the WG's ability to represent extremes the threshold weighted CRPS (twCRPS) is explored. For three different thresholds the twCRPS corroborates the results for the CRPS. The use of proper scoring rules in conjunction with a DM test is highlighted as a powerful tool for assessing WG skill.

## 1 Introduction

Statistical distributions are widely used to downscale climate projections for impact analysis (Maraun and Widmann, 2018). Current projection methods of future climate impacts rely on General Circulation Models (GCMs), with model intercomparison projects like the Coupled Model Intercomparison Project Phase 6 (CMIP6) providing a multi-model ensemble of climate projections (Eyring et al., 2016). These projections lack the resolution to resolve local weather resulting in the need to use observations to downscale GCMs. A weather generator (WG) is a statistical distribution trained on local weather that generates





random variables that mimic a local weather variable, often taking large scale climate variables as input (Maraun et al., 2010). A WG therefore links local weather to large scale climate and can then be forced by a GCM to project into the future.

A high priority variable for downscaling is precipitation due to its relevance to impacts and the importance of sub grid scale processes in its occurrence. A variety of methods for downscaling precipitation are available, for a review of these see Maraun et al. (2010) and Maraun and Widmann (2018). A large intercomparison of methods can be found in Gutiérrez et al. (2019).

Two WG methods are relevant to this study: weather type WGs and Generalised Linear Model (GLM) WGs.

A typical weather types WG approach is as follows. Weather types are created using clustering algorithms on the spatial patterns of large scale atmospheric variables (e.g. Bogardi et al., 1993; Vrac et al., 2007; Garavaglia et al., 2010). This provides a gridded dataset of discrete weather types. Given these weather types, the empirical distribution of local precipitation data can be sampled from, or another WG method could be used. San-Martín et al. (2017) conditioned a GLM WG on weather types

created using k-means clustering. The advantage of using weather types is that they summarise large scale circulation patterns, which are of significant physical relevance to precipitation. However, if the goal is to use weather types for GCM downscaling, a correspondence between weather types generated for observations and weather types generated for GCMs is needed. For clustering algorithms either the observation and GCM time series need to be concatenated and clustering performed over the whole set, or clustering needs to be done on the observed set and a classification algorithm used to classify patterns in the

GCMs (e.g. k-nearest neighbours; Moron et al., 2008).

GLM WGs have been used frequently in downscaling studies (Chandler and Wheater, 2002; Yang et al., 2005; Furrer and Katz, 2007). Precipitation has a highly skewed distribution, so linear regression performs poorly (Gutiérrez et al., 2019). However relationships between atmospheric variables and precipitation are available and these can be fitted using the more flexible GLM method. A GLM WG fits a relationship between local scale precipitation data, for example from weather stations,

to large scale atmospheric variables.

In this study a new alternative to clustering-based weather types is proposed. Here the weather types used, hereafter referred to as storm types, are based on physical a priori information about storms. The storm types (Catto and Dowdy, 2023) are an updated set of the ones used in Catto and Dowdy (2021) and provide a discrete classification of when a cyclone, thunderstorm or front is present in a grid cell. These types are labelled using thresholds in atmospheric variables from the European Centre

for Medium-Range Weather Forecasts (ECMWF) Reanalysis version 5 (ERA5) (Hersbach et al., 2020). The main advantage of using these classifications is that they are based on meterological processes and are therefore physically interpretable.

The WG proposed by this study is a Bayesian GLM WG conditioned on the a priori storm types. Here the WG is trained on large scale climate variables from ERA5 and on local precipitation data from the Met Office MIDAS Open: UK Land Surface Stations Data (Met Office, 2023). Throughout this study the large scale atmospheric variables are referred to as the covariates

of the WG and the local scale variable as local precipitation or just precipitation. The WG follows a standard approach (Katz, 1999; Husak et al., 2007; Furrer and Katz, 2007), modelling occurrence and amounts separately, and developing on previous work by using a Bayesian framework with informative priors. Other examples of Bayesian WGs use weakly informative priors (e.g. Verdin et al., 2019) or uniform priors (e.g. Legasa and Gutiérrez, 2020) but here the prior distributions are based on the Prior Predictive Distribution (PrPD), i.e. based on the observed predicted distribution for precipitation, before looking at the





data. This means prior choices are made using the observable quantity of precipitation rather than beliefs about the unobserved regression parameters.

Scoring rules compare observations to predictive probability distributions. A proper scoring rule has the lowest expected score if the predicted distribution equals the underlying distribution of the observation. Proper scoring rules are often used for the evaluation of weather forecasts (Gneiting et al., 2007) and have been used infrequently to evaluate WGs (Rahmani

et al., 2016; Sohrabi and Brissette, 2021; Krouma et al., 2022). As proper scoring rules are a probabilistic metric they are ideal for WG evaluation when the WG can be considered a predictive probabilistic distribution. This study aims to provide a clear example of the use of proper scoring rules in the context of weather generators, by quantifying the improvement to the WG from the inclusion of storm types. Further, a key factor that is rarely considered in the application to WGs, is the statistical significance of metrics used to rate WGs which is addressed here using Diebold-Mariano (DM) significance tests.

Sect. 2 outlines the methods and data used, including the atmospheric and station data (Sect. 2.1), storm types (Sect. 2.2), GLM (Sect. 2.3), the covariate choice for the different WG models (Sect. 2.4), the prior choice for the Bayesian WG model (Sect. 2.5), the scoring rules used to evaluate the model (Sect. 2.6), the Diebold-Mariano test (Sect. 2.7), calibration assessment tools (Sect. 2.8) and how the WG is implemented (Sect. 2.9). Sect. 3 outlines the results of the model comparison, first presenting the prior predictive distribution used to inform the prior choice (Sect. 3.1), then the posterior distribution of the

GLM regression parameters (Sect. 3.2), then comparing the calibration of the models (Sect. 3.3), and comparing the models using scoring rules (Sect. 3.4). Finally Sect. 4 is a discussion and conclusion.

## 2  Methods and Data

### 2.1  Data

Atmospheric data is from the European Centre for Medium-Range Weather Forecasts (ECMWF) Reanalysis version 5 (ERA5)

(Hersbach et al., 2020). The atmospheric variables are instantaneous and selected at 6 hourly intervals, at the grid cells of the UK weather stations used and the time period of January 1980 to December 2018 inclusive. The spatial resolution of ERA5 is a $0.25° \times 0.25°$ regular latitude longitude grid. Three different atmospheric variables are chosen as covariates for the model: vertical velocity at 500hPa, relative humidity at 850hPa and specific humidity at 850hPa. Covariate choice is discussed in Sect. 2.4. Precipitation station data is from the Met Office MIDAS Open: UK Land Surface Stations Data. Three different MIDAS

Open weather stations are used: station 01336 in Plymouth-Mountbatten in Devon, station 18974 in Tiree off the west coast of mainland Scotland, and station 00440 in Wattisham in Suffolk. These stations are chosen to span different climatologies around the UK and because they have hourly datasets that spanned the majority of the time domain being considered (January 1980 to December 2018 inclusive). The Tiree and Wattisham stations have a resolution of $0.2 \text{ mm hr}^{-1}$ with periods where the resolution increased to $0.1 \text{ mm hr}^{-1}$. The Plymouth station only has a resolution of $0.2 \text{ mm hr}^{-1}$. For consistency the weather

station precipitation data is rounded to the nearest $0.2 \text{ mm hr}^{-1}$. It is then resampled to 6 hourly accumulations to match the temporal resolution of the storm types dataset.



**Table 1.** Occurrence fractions (number of 6 hour time steps when precipitation was greater than 1 mm 6h$^{-1}$ divided by the total number of time steps) for the three different stations and 8 storm types (including no storm) and the overall occurrence fraction. Note these are the occurrence fractions for each storm type with respect to the number of occurrences of that storm type and therefore summing over the different storm types should not be expected to equal 1.

| | Occurrence fraction | | | | | | | | |
|---|---|---|---|---|---|---|---|---|---|
| Station | Overall | N | CO | FO | TO | CF | CT | FT | CFT |
| Plymouth | 0.14 | 0.04 | 0.17 | 0.16 | 0.13 | 0.39 | 0.28 | 0.20 | 0.40 |
| Tiree | 0.20 | 0.06 | 0.20 | 0.19 | 0.23 | 0.40 | 0.32 | 0.31 | 0.42 |
| Wattisham | 0.10 | 0.02 | 0.12 | 0.10 | 0.09 | 0.29 | 0.18 | 0.16 | 0.35 |

## 2.2 Storm types

The European storm types dataset (Catto and Dowdy, 2023) contains a discrete set of weather types, based on ERA5 atmospheric data. Cyclones are identified as the area containing a minima of Mean Sea Level Pressure (MSLP) enclosed by the
outermost MSLP contour (method from Wernli and Schwierz (2006)). Fronts are identified using the Thermal Front Parameter (TFP) which is calculated using the gradient of 850hPa thermal wet bulb potential. Fronts are identified as contour lines where the gradient of the TFP is zero within a baroclinic zone (Sansom and Catto, 2022). Thunderstorms are identified using a proxy based on Convective Available Potential Energy (CAPE), bulk wind sheer from 0 to 6km, total totals index and the Laplacian of 500hPa geopotential height (Dowdy and Brown, 2023). Thresholds are based on lightning observations (Dowdy, 2020). The
dataset is 6 hourly and available from January 1980 to December 2018 inclusive. The dataset has the same $0.25° \times 0.25°$ grid as the ERA5 atmospheric data.

The set is made up of 7 different storm types for each combination of storm present: cyclone only (CO), thunderstorm only (TO), front only (FO), cyclone and front (CF), front and thunderstorm (FT), cyclone and thunderstorm (CT) and cyclone, front and thunderstorm (CFT). Additionally there could be no storm present (N). Fig. 1 shows the distribution of 6 hourly
precipitation amounts values (greater than 1 mm 6h$^{-1}$) and Table 1 shows the fraction of time steps when precipitation was greater than 1 mm 6h$^{-1}$ for the different storm types at the three stations.

## 2.3 Generalised linear model weather generator

A Generalised Linear Model (GLM) approach was chosen for the weather generator. Precipitation data contains a large number of zeros. As such, occurrence and amounts of precipitation are modelled separately with amounts conditional on occurrence
and otherwise set to 0 mm 6h$^{-1}$. This is an example of a hurdle model which has been frequently used in epidemiology and public health research (Feng, 2021), where count data often contains zeros generated by a different process to the non zero counts. A hurdle is modelled discretely and the amount it is passed by is modelled by a distribution conditional on the hurdle being passed. The physical processes determining occurrence and amounts are different, as such the covariates are not



**Figure 1.** Distribution of precipitation amount greater than $1\text{mm}\,6\text{h}^{-1}$ for the different storm types for the three stations. Note that the y axis is density with respect to each station and not the relative counts of precipitation values for each station; e.g. in Tiree there is more occurrence of precipitation, and therefore more precipitation on average, but its distribution shown has a similar height to the other stations. Kernel density estimates are made using the logKDE package (Jones et al., 2018).




necessarily the same for occurrence and amounts (Wilby and Wigley, 2000). Here precipitation occurrence is modelled using

logistic regression and amounts given occurrence modelled by a Gamma GLM. This is a standard approach and can be found in Chandler and Wheater (2002); Yang et al. (2005); Furrer and Katz (2007); Ambrosino et al. (2014). Occurrence is defined as a precipitation amount greater than $1$ mm $6\text{h}^{-1}$.

Let $r_t$ be the amount of precipitation in a 6 hour time step $t$, where any precipitation amount less than or equal to the threshold of occurrence, $r_{\text{th}} = 1$ mm $6\text{h}^{-1}$, is considered dry and set to zero. Then let $o_t$ be an indicator variable for the occurrence of

precipitation, so $o_t$ is 0 (1) if $r_t \leq 1$ mm $6\text{h}^{-1}$ ($r_t > 1$ mm $6\text{h}^{-1}$). Finally, let $y_t$ be the amount of precipitation greater than $r_{\text{th}}$ in a 6 hour time step $t$ given the occurrence of precipitation, i.e. $o_t = 1$. The model is then specified as follows:

$$o_t | p_t \sim \text{Bernoulli}(p_t), \tag{1}$$

$$y_t | \alpha, \beta_t \sim \text{Gamma}(\alpha, \beta_t), \tag{2}$$
$$\alpha/\beta_t = \mu_t.$$


$$r_t = \begin{cases} y_t + r_{\text{th}} & \text{if } o_t = 1, \\ 0 & \text{if } o_t = 0, \end{cases} \tag{3}$$

$$= o_t(y_t + r_{\text{th}}). \tag{4}$$

The occurrence of precipitation, $o_t$, is modelled as a Bernoulli trial with probability of occurrence $p_t$. This is referred to as the occurrence model. The amount of precipitation given precipitation occurrence, $y_t$, is modelled as a Gamma distributed

variable with mean $\mu_t$ and constant shape parameter $\alpha$. This is referred to as the amounts model. The combination of these two models to simulate $r_t$ is referred to as the overall model. The quantities $p_t$ and $\mu_t$ are related to a linear combination of covariate values via a logit and logarithmic link function respectively:

$$g(p_t) = \log\left(\frac{p_t}{1 - p_t}\right) = \boldsymbol{x}_{t,o}^T \boldsymbol{\theta}_o^{(s)}, \tag{5}$$

$$g(\mu_t) = \log(\mu_t) = \boldsymbol{x}_{t,a}^T \boldsymbol{\theta}_a^{(s)}. \tag{6}$$

The vector $\boldsymbol{x}_{t,\cdot}$ is a column vector of covariates at time step $t$, $\boldsymbol{x}_{t,\cdot} = (1, x_1, x_2, ..., x_n)_{t,\cdot}^T$. The vector $\boldsymbol{\theta}_{\cdot}^{(s)}$ is a column vector of regression parameters, $\boldsymbol{\theta}_{\cdot}^{(s)} = (\theta_0, \theta_1, \theta_2, ..., \theta_n)_{\cdot}^{(s)T}$. For $\boldsymbol{x}_{t,\cdot}$ and $\boldsymbol{\theta}_{\cdot}^{(s)}$ the subscript $o$ and $a$ denote if the covariates and regression parameter are for the occurrence or amounts models respectively. There is no reason to expect the relationship between discrete storm types and precipitation to be linear, and so the storm types are not included as additional covariates.

Instead, a separate set of regression parameters, $\boldsymbol{\theta}_{\cdot}^{(s)}$, is fitted for each of the storm types $s$. For models without storm types, which are fitted as benchmarks for comparison, the regression parameters are fitted over the whole dataset, i.e. superscript $s$ can be ignored. The covariate choice is discussed in more detail in Sect. 2.4.



The GLM is treated within a Bayesian framework (Gelman et al., 2013). Equations 1 and 2 are the distributions of $o_t$ and $y_t$ given the regression parameters, $\boldsymbol{\theta}_o^{(s)}$ and $\boldsymbol{\theta}_a^{(s)}$ respectively, which are linked to these distributions by equations 5 and 6 respectively. The occurrence and amounts models then have the standard Binomial and Gamma densities as likelihoods:

$$\pi(\boldsymbol{o}|\boldsymbol{\theta}_o^{(s)}) = \prod_{t=1}^{T} p_t^{o_t}(1-p_t)^{(1-o_t)}, \tag{7}$$

$$\pi(\boldsymbol{y}|\boldsymbol{\theta}_a^{(s)}) = \prod_{t=1}^{T} \frac{\beta_t^{\alpha}}{\Gamma(\alpha)} y_t^{\alpha-1} e^{-\beta_t y_t}. \tag{8}$$

The amounts model is conditionally independent of the occurrence model and so the joint posterior distribution for $\boldsymbol{\theta}_a^{(s)}$ and $\boldsymbol{\theta}_o^{(s)}$ can be written in terms of the posterior distributions of the amounts and occurrence models as:

$$\pi(\boldsymbol{\theta}_o^{(s)}, \boldsymbol{\theta}_a^{(s)}|\boldsymbol{r}) = \pi(\boldsymbol{\theta}_a^{(s)}|\boldsymbol{y})\pi(\boldsymbol{\theta}_o^{(s)}|\boldsymbol{o}). \tag{9}$$

The posterior distribution for the regression parameters is proportional by Bayes' theorem to the likelihood times the prior over the regression parameters, and so for the occurrence and amounts models is:

$$\pi(\boldsymbol{\theta}_o^{(s)}|\boldsymbol{o}) \propto \pi(\boldsymbol{o}|\boldsymbol{\theta}_o^{(s)})\pi(\boldsymbol{\theta}_o^{(s)}), \tag{10}$$

$$\pi(\boldsymbol{\theta}_a^{(s)}|\boldsymbol{y}) \propto \pi(\boldsymbol{y}|\boldsymbol{\theta}_a^{(s)})\pi(\boldsymbol{\theta}_a^{(s)}). \tag{11}$$

We want a prediction of the amount of precipitation given the covariates at a time step $t$ which is provided by the posterior predictive distribution (PPD), $\pi(r_t^*|\boldsymbol{r})$ where $r_t^*$ is the prediction for precipitation. The PPD for $r_t^*$ can be written in terms of the PPD for $o_t^*$ and $y_t^*$, the predicted values of $o_t$ and $y_t$:

$$\pi(r_t^*|\boldsymbol{r}) = \begin{cases} \pi(o_t^*=1|\boldsymbol{o})\pi(y_t^*|\boldsymbol{y}) & \text{if } r_t^* > 1 \text{ mm 6h}^{-1}, \\ \pi(o_t^*=0|\boldsymbol{o}) & \text{if } r_t^* = 0 \text{ mm 6h}^{-1}. \end{cases} \tag{12}$$

The PPD for some new prediction $d^*$ given data $\boldsymbol{d}$ with a vector of parameters $\boldsymbol{\theta}$ is found by marginalising the distribution of the new prediction $d^*$ given $\boldsymbol{\theta}$ over the posterior distribution of $\boldsymbol{\theta}$:

$$\pi(d^*|\boldsymbol{d}) = \int_{\Theta} \pi(d^*|\boldsymbol{\theta})\pi(\boldsymbol{\theta}|\boldsymbol{d})d\boldsymbol{\theta}. \tag{13}$$

Sect. 2.9 contains details on how the Bayesian GLM is implemented.

## 2.4 Covariates

Three different covariate choices are used; intercept only, simple seasonal variables and atmospheric variables. These can be seen in Table 2 and will be referred to as the Null model, Seasonal model and Atmospheric model. The seasonal covariates are $\sin\left(2\pi\frac{\text{DOY}}{365}\right)$ and $\cos\left(2\pi\frac{\text{DOY}}{365}\right)$. A different set of atmospheric variables are used for precipitation amounts versus occurrence.





**Table 2.** Table of covariates used. The atmospheric variables used are vertical velocity in pressure coordinates w /Pa s$^{-1}$, relative humidity r /%, and specific humidity q /kg kg$^{-1}$ at 500, 850 and 850 hPa pressure levels respectively.

| | Covariates | |
| --- | --- | --- |
| Model | Occurrence | Amounts |
| Null | Intercept | Intercept |
| Seasonal | Intercept, $\sin\left(2\pi\frac{\text{DOY}}{365}\right)$, $\cos\left(2\pi\frac{\text{DOY}}{365}\right)$ | Intercept, $\sin\left(2\pi\frac{\text{DOY}}{365}\right)$, $\cos\left(2\pi\frac{\text{DOY}}{365}\right)$ |
| Atmospheric | Intercept, $\sin\left(2\pi\frac{\text{DOY}}{365}\right)$, $\cos\left(2\pi\frac{\text{DOY}}{365}\right)$, w500, r850 | Intercept, $\sin\left(2\pi\frac{\text{DOY}}{365}\right)$, $\cos\left(2\pi\frac{\text{DOY}}{365}\right)$, w500, q850 |

As outlined in Maraun and Widmann (2018), the choice of large scale atmospheric covariates for precipitation weather generation should have a physical relationship between each covariate and precipitation and the covariate should be well correlated with precipitation. Vertical velocity at 500hPa (w500) was chosen for both occurrence and amounts, relative humidity at 850hPa (r850) chosen for occurrence, and specific humidity at 850hPa (q850) chosen for amounts. These vertical pressure levels are chosen as 850hPa is approximately the cloud base and 500hPa is higher up where where we expect to see strong ascent when precipitation occurs (e.g. Catto et al., 2010). The atmospheric covariates chosen here are taken from Table 11.3 Chapter 11 of Maraun and Widmann (2018). Vertical velocity is related to both occurrence and amounts due to the relationship between the ascent of moist air and cloud formation and precipitation. Relative humidity is related to occurrence due to its relevance in cloud formation. The greater the specific humidity the more water vapor is available, relating to amounts.

The distribution of precipitation versus w500 (Fig. 2a) shows a relationship of greater precipitation for more negative w500 i.e. faster ascent. The distribution of precipitation versus q850 (Fig. 2b) shows a relationship where the greater q850 and therefore more water vapor, the greater the amount of precipitation. The different distributions of w500 and r850 for occurrence and non-occurrence of precipitation can be seen in Figs. 2c and 2d. For w500 occurrence happens when w500 is more negative, which indicates ascent, and for r850 cloud formation occurs in air when relative humidity is greater than 100%. It should be noted that the covariates are area averages over an ERA5 grid cell and only taken at one pressure level not the entire air column. This means that precipitation can occur when w500 is positive and r850 is less than 100%, it is just more likely on average when w500 is more negative and r850 is closer to 100%.

## 2.5 Priors

A prior distribution is needed for all regression parameters and for the amounts Gamma GLM models a prior is also needed for the shape parameter. The choice of prior distribution describes our expectation for the parameters before seeing the data and fitting the model. Here normal priors with mean zero are chosen for all regression parameters including the intercept, so the prior choice is symmetric as to if the parameters are positive or negative. For the models with storm types, each regression parameter is given the same prior regardless of storm type, but not necessarily the same prior choices as the equivalent model





**Figure 2.** ERA5 atmospheric data in the grid cell above the Wattisham MIDAS Open weather station. (a) The two distributions of vertical velocity in pressure coordinates at 500hPa (w500) for precipitation occurrence (precip) and non occurrence (dry). (b) The two distributions of relative humidity at 850hPa (r850) for precipitation occurrence and non-occurrence. (c) Vertical velocity in pressure coordinates at 500hPa (w500) plotted against precipitation amount greater than 1mm 6h$^{-1}$. (d) Specific humidity plotted against precipitation amount greater than 1mm 6h$^{-1}$.





without storm types. Similarly common regression parameters (e.g. the intercept) for the different Null, Seasonal and Atmospheric models, are not assumed to have the same prior distribution. For the shape parameter a Gamma prior is used as the shape parameter is positive and its value is not expected to be much greater than zero.

Before fitting the models to precipitation data it is difficult to know what the consequence of different parameter values will
be on the final precipitation distribution. In order to explore the consequence of different prior choices, the Prior Predictive Distribution (PrPD) is used (Equations 14 and 15 for amounts and occurrence respectively).

$$\pi(y^*) = \int_\Theta \pi(\boldsymbol{y}^*|\boldsymbol{\theta}_y^{(s)})\pi(\theta)d\theta \tag{14}$$

$$\pi(o^*) = \int_\Theta \pi(\boldsymbol{o}^*|\boldsymbol{\theta}_o^{(s)})\pi(\theta)d\theta \tag{15}$$

The PrPD is similar to the PPD, but is the likelihood marginalised over the prior distribution rather than the posterior distribution.

### 2.6 Scoring rules

For model comparison proper scoring rules are used to quantify the predictive skill of the weather generator. Proper scoring rules are a common tool to evaluate forecasts against observations (Gneiting et al., 2007; Benedetti, 2010; Gneiting and
Katzfuss, 2014). The scoring rules used to evaluate the weather generator are the Brier Score (BS), the Continuous Ranked Probability Score (CRPS) and the threshold weighted Continuous Ranked Probability Score (twCRPS). The BS is used to score the occurrence distribution, CRPS the overall and amounts distributions, and the twCRPS the overall distribution to focus on the tails of the predictive distribution.

The BS is defined as:

$$BS = \frac{1}{T}\sum_{t=1}^{T}(p_t^* - o_t)^2, \tag{16}$$

where $o_t$ is the observed occurrence as defined in Sect. 2.3 and $p_t^*$ is the posterior predictive probability of occurrence ($o_t^* = 1$) given the data ($\boldsymbol{o}$), $p_t^* = \pi(o_t^* = 1|\boldsymbol{o})$. In terms of independent random samples of the PPD, $o_{t,i}^*$ ($i = 1,...,N$), $p_t^*$ is the sample mean: $p_t^* = 1/N\sum_{i=1}^{N} o_{t,i}^*$.

The CRPS compares a probability distribution to an observation. Defining a random variable $X$ with cumulative distribution
function (CDF) $F(x)$, which is the predicted distribution for observation $z$ at time $t$ the CRPS is defined as:

$$\mathrm{CRPS}(F(\cdot), z) = \int_{-\infty}^{\infty} (F(x) - H(x - z))^2 dx, \tag{17}$$




Where $H(x-z)$ is a Heaviside step function at the observation $z$ ($H(X-z) = 1$ if $X \geq z$ and 0 otherwise). The CRPS can be expressed in terms of the expectations of independent random samples, $x$ and $x'$, of the predictive distribution:

$$\text{CRPS}(F(\cdot), z) = E_{X \sim F}[|X - z|] - \frac{1}{2} E_{X \sim F}[|X - X'|], \tag{18}$$

In this study $F(X)$ is the CDF of the WG PPD at time $t$ and the CRPS can then be found from samples of the amounts and overall models. These CRPS values are then averaged over time to provide a score for each WG model.

The twCRPS is a modification to the CRPS, including a weight function, $w(X)$, under the integral (Gneiting and Ranjan, 2011):

$$\text{twCRPS}(F(\cdot), z) = \int_{-\infty}^{\infty} (F(x) - H(X - z))^2 w(x) dx. \tag{19}$$

The twCRPS has an equivalent form to Equation 18 (Taillardat et al., 2023):

$$\text{twCRPS}(F(\cdot), z) = E_{X \sim F}[|v(X) - v(z)|] - \frac{1}{2} E_{X \sim F}[|v(X) - v(X')|], \tag{20}$$

where,

$$v(X) - v(X') = \int_{X'}^{X} w(\zeta) d\zeta. \tag{21}$$

Using a Normal cumulative distribution as the weight function (e.g. Gneiting and Ranjan (2011)) the function $v(X)$ is:

$$v(X) = (X - \mu)\Phi_{\mu,\sigma}(X) + \sigma^2 \varphi_{\mu,\sigma}(X), \tag{22}$$

where $\varphi_{\mu,\sigma}(X)$ is the normal density function.

For the CRPS, twCRPS and BS a better score is a score closer to zero. For the BS a perfect score would have a PPD probability of occurrence of one for all occurrences and zero for all non-occurrences. Considering Equation 16 this means $p_t^* = o_t$ for all $t$ and $BS = 0$. The CRPS is the area under the squared difference between the CDF of the model and a

Heaviside step function at the observation (Equation 17). A perfect CRPS of zero would occur when the CDF of the model is equal to the step function at the observation; i.e. all the probability density is at the observation. The CRPS scores how close the CDF is to a step function at the observation.

## 2.7 Diebold-Mariano test

To determine if the average score differences are statistically significant a Diebold-Mariano (DM) test is performed (Diebold

and Mariano, 1995). Diebold and Mariano (1995) propose a test for a sequence of loss differentials (differences in loss functions between two models). Here this test is used on a sequence of score differentials; the difference between the scores of two different models compared with the same observations. Let $\Delta_t = S_{1,t} - S_{2,t}$ be the score difference between two different WG





models at time $t$. Assume that the sequence is covariance stationary. This means the score differential sequence has constant mean $\mu_{\text{DM}}$ and variance $\sigma^2_{\text{DM}}$ and a constant autocovariance function $\gamma(\tau)$ for lag $\tau$:

$$E(\Delta_t) = \mu_{\text{DM}},$$

$$\text{cov}(\Delta_t, \Delta_{t-\tau}) = \gamma(\tau), \tag{23}$$

$$0 < \text{var}(\Delta_t) = \sigma^2_{\text{DM}} < \infty.$$

Given these assumptions the sample mean score difference, $\overline{\Delta} = \sum_{t=0}^{T} \Delta_t$, will tend towards a normal distribution with mean $\mu_{\text{DM}}$ as sample size increases (Diebold, 2015). A $\mu_{\text{DM}}$ of 0 would correspond to the hypothesis of equal score. Therefore, under these assumptions, under the null hypothesis of equal scores:

$$\text{DM} = \frac{\overline{\Delta}}{\hat{\sigma}_{\text{DM}}} \rightarrow N(0,1), \tag{24}$$

where $\hat{\sigma}_{\text{DM}}$ is a consistent estimate of the standard deviation of $\overline{\Delta}$. Crucially autocorrelation between score differences at different time steps needs to be taken into account. Here this is done by using the sample variance divided by the effective sample size ($N_{\text{eff}}$) of the sequence of score differences:

$$\hat{\sigma}^2_{DM} = \frac{1}{N_{\text{eff}}T} \sum_{t=0}^{T} (\Delta_t - \overline{\Delta})^2, \tag{25}$$

with $N_{\text{eff}}$ defined as:

$$N_{\text{eff}} = \frac{T}{1 + 2\sum_{k=1}^{\infty} \rho(k)}, \tag{26}$$

Where $\rho(k)$ is the autocorrelation at lag $k$. The R function acf is used to estimate $\rho(k)$ and the sum in Equation 26 is truncated once $\rho(k) < 0.005$. The DM test statistic, under the assumptions in Equation 23, can be used in a hypothesis test for the case of equal score by comparison to Normal quantiles.

## 2.8 Calibration and sharpness

Calibration refers to the "statistical consistency" (Gneiting et al., 2007) between the predictive distribution and the observations. Scoring rules quantify both calibration and sharpness (Gneiting et al., 2007), where a sharper probability distribution has less spread. This is desirable as a sharper probability distribution implies a reduction in internal variability for precipitation given the input covariates.

To visualise just calibration two methods are used: the Probability Integral Transform (PIT) histogram and comparing observations to the PPD directly. Let $F_t(r)$ be the CDF of the WG at time $t$. The PIT histogram is the histogram of the CDF values of the WG at the observations, $r_t$, at each time $t$:

$$\text{PIT} = F_t(r_t). \tag{27}$$



A well calibrated predictive model will have a uniformly distributed PIT histogram between 0 and 1 (Rosenblatt, 1952). In addition to the PIT histogram is it useful to compare draws from the PPD to the probability density function (PDF) of the
observations over the whole time series (using kernel density estimation). This equates to a comparison between the WG 'climate' (long term precipitation average) and the observed climate.

## 2.9 Implementation

The intended output of a precipitation WG is a time series of precipitation values. Here this means samples from the PPD of the WG, $\pi(r_t^*|\mathbf{r})$. The ability to make probabilistic statements about the PPD is also needed, requiring its integration. The
calculation of CRPS and twCRPS scores for the amounts and overall PPDs again requires an integral. Finally viewing the distributions of the PPD, PrPD and the posterior of the regression parameters is needed. These integrals cannot be solved analytically but can be estimated using sampling.

As is typical of Bayesian inference this sampling is done using a Markov chain Monte Carlo (MCMC) algorithm (Gelman et al., 2013). Samples of the posterior distributions $\pi(\boldsymbol{\theta}_o^{(s)}|\mathbf{o})$ and $\pi(\boldsymbol{\theta}_a^{(s)}|\mathbf{y})$ are generated using Hamiltonian Monte Carlo
(HMC) (Gelman et al., 2013) using the R package brms (Bürkner, 2017; Stan Development Team, 2024), which efficiently samples the distribution using the time evolution of the Hamiltonian equations of a particle in a potential well. For each posterior sample from the occurrence $\boldsymbol{\theta}_{o,i}^{(s)}$ and amounts $\boldsymbol{\theta}_{a,i}^{(s)}$ model, a sample is drawn from the likelihood given that sample:

$$
\begin{aligned}
\boldsymbol{\theta}_{o,i}^{(s)} &\sim \pi(\boldsymbol{\theta}_o^{(s)}|\mathbf{o}), \\
o_{t,i}^* &\sim \pi(o_t^*|\boldsymbol{\theta}_{o,i}^{(s)}),
\end{aligned}
\tag{28}
$$

$$
\begin{aligned}
\boldsymbol{\theta}_{a,i}^{(s)} &\sim \pi(\boldsymbol{\theta}_a^{(s)}|\mathbf{y}), \\
y_{t,i}^* &\sim \pi(y_t^*|\boldsymbol{\theta}_{a,i}^{(s)}).
\end{aligned}
\tag{29}
$$

As $o_t^*$ is an indicator variable for occurrence, samples from the overall PPD, $\pi(r_t^*|\mathbf{r})$, are straightforwardly generated by multiplying samples from the two models and adding the occurrence threshold ($r_{\mathrm{th}} = 1$ mm 6h$^{-1}$): $r_{t,i}^* = o_{t,i}^*(y_{t,i}^* + r_{\mathrm{th}})$.

The different covariate choices result in six different models for the WG; Null, Seasonal and Atmospheric without storm types and the same with storm types. Models with greater complexity will have a greater ability to fit to data simply as a result
of the higher number of parameters. To prevent overfitting, the WG models are fitted using 10 fold cross validation. Each time step is assigned a random fold and the data assigned to a fold is excluded when fitting the model for that fold. Let $\mathbf{o}_k$ and $\mathbf{y}_k$ be the set of observations in fold $k$ assigned to time step $t$ and $\mathbf{o}_{-k}$ and $\mathbf{y}_{-k}$ be the set of observations not in fold $k$. Then,

$$
\begin{aligned}
\boldsymbol{\theta}_{o,k,i}^{(s)} &\sim \pi(\boldsymbol{\theta}_o^{(s)}|\mathbf{o}_{-k}), \\
o_{t,i}^* &\sim \pi(o_t^*|\boldsymbol{\theta}_{o,k,i}^{(s)}),
\end{aligned}
\tag{30}
$$

$$
\begin{aligned}
\boldsymbol{\theta}_{a,k,i}^{(s)} &\sim \pi(\boldsymbol{\theta}_a^{(s)}|\mathbf{y}_{-k}), \\
y_{t,i}^* &\sim \pi(y_t^*|\boldsymbol{\theta}_{a,k,i}^{(s)}).
\end{aligned}
\tag{31}
$$




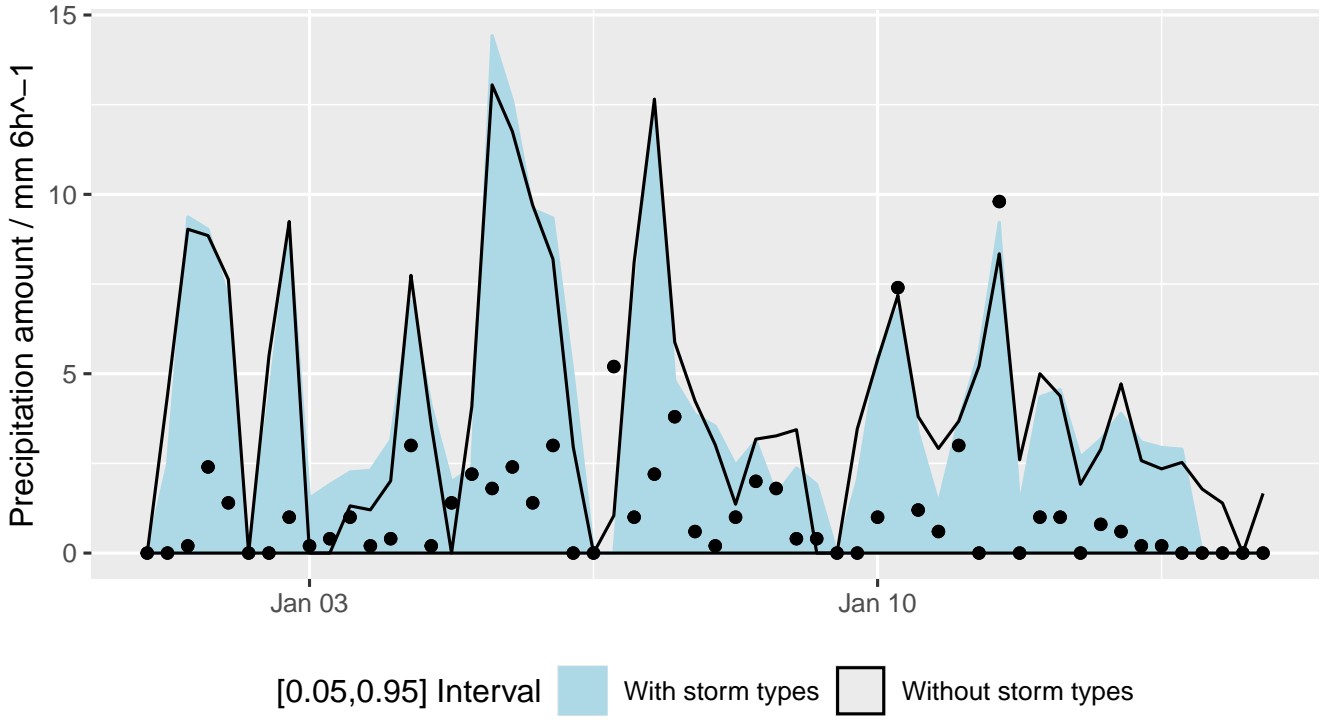

**Figure 3.** Time series (1st to 15th of January 2000) of 6 hourly precipitation for the Tiree station. The shaded areas are the 5% to 95% interval for the atmospheric covariate weather generator conditions with storm types and without storm types. The ● are the observed precipitation amounts.

For each fold $k$ the WG models are trained using observations $o_{-k}$ and $y_{-k}$ and are validated using proper scoring rules and calibration tests on the WG predictions for observations $o_k$ and $y_k$.

## 3 Results

Fig. 3 shows a time series of precipitation for the Tiree station along with the Atmospheric model with storm types and without storm types. The WG model does not predict precipitation on a time step by time step basis. Instead it predicts different width distributions depending on the covariates. This results in time steps when the WG distribution is restricted to be close to 0, i.e. high precipitation is unlikely, and time steps when the WG distribution is wider and higher precipitation is given greater chance. The two models in Fig. 3 have similar distributions highlighting the need for a quantitative method of comparison.





**Table 3.** The standard deviations of the normal, mean zero, prior distributions for regression parameters for the occurrence and amounts models. Note in brackets the standard deviation of the prior distribution for the intercept of the occurrence model with storm types, which is the only difference between the prior choice for the models with and without storm types.

| | Regression parameter prior standard deviation | |
| --- | --- | --- |
| Covariate | Occurrence | Amounts |
| Intercept | 2 (10 with storm types) | 1 |
| $\sin\left(\frac{\text{DOY}}{365}\right), \cos\left(\frac{\text{DOY}}{365}\right)$ | 1 | 0.2 |
| w500 | 2 Pa s$^{-1}$ | 1 Pa s$^{-1}$ |
| r850 | 0.02% | - |
| q850 | - | 100 kg kg$^{-1}$ |

## 3.1 Prior Predictive Distribution (PrPD)

As discussed in Sect. 2.5 the priors for the Null, Seasonal and Atmospheric occurrence and amounts models with and without storm types need to be chosen. Specifically, the priors over the regression parameters are mean 0 normal distributions, and the prior over the shape parameter for the Gamma GLM amounts models is a Gamma distribution. The standard deviations of the regression parameter normal distributions and the rate and shape parameter for the shape parameter Gamma prior are chosen by examining the PrPD.

Draws from the PrPD are summarised in the following way. For the occurrence model the histogram of time averaged occurrence $\bar{o} = \frac{1}{T}\sum_{t=1}^{T} o_t$ is used. Priors are chosen where the distribution of $\bar{o}$ covers values between 0.1 and 0.3; i.e. reasonable $\bar{o}$ values are not being excluded. Fig. 4 shows this histogram for the Atmospheric model with and without storm types for the Wattisham station. For the amounts model draws of the PrPD can be looked at directly. Priors are chosen so that draws from the PrPD are over a spread of precipitation values and not strongly favouring very large or very small precipitation values.

This can be seen in Fig. 5 again for the Atmospheric model for Wattisham. The prior standard deviations for the regression parameters can be seen in Table 3. The shape and rate parameter for the Gamma prior are chosen to be 2 and 1 respectively. The observed precipitation time averaged occurrence and amounts distribution are marked in black on these figures for reference, but the purpose of this step is not to match the PrPD to the data. See Figs. S4–S21 in the supplement for plots of the PrPD for all different stations and models.

The final prior choice is the same for the covariates across the Null, Seasonal and Atmospheric models. For example, the Normal$(0,1)$ prior for the intercept of the amounts model is the same with and without additional covariates, despite the knowledge that the intercept would have a different value, and physical interpretation, for each model. However, following checking the PrPD, the final prior choice for the occurrence model with storm types required a larger normal standard deviation to the Atmospheric model without storm types (10 versus 2) for the PrPD to cover a reasonable range of occurrences. This is the





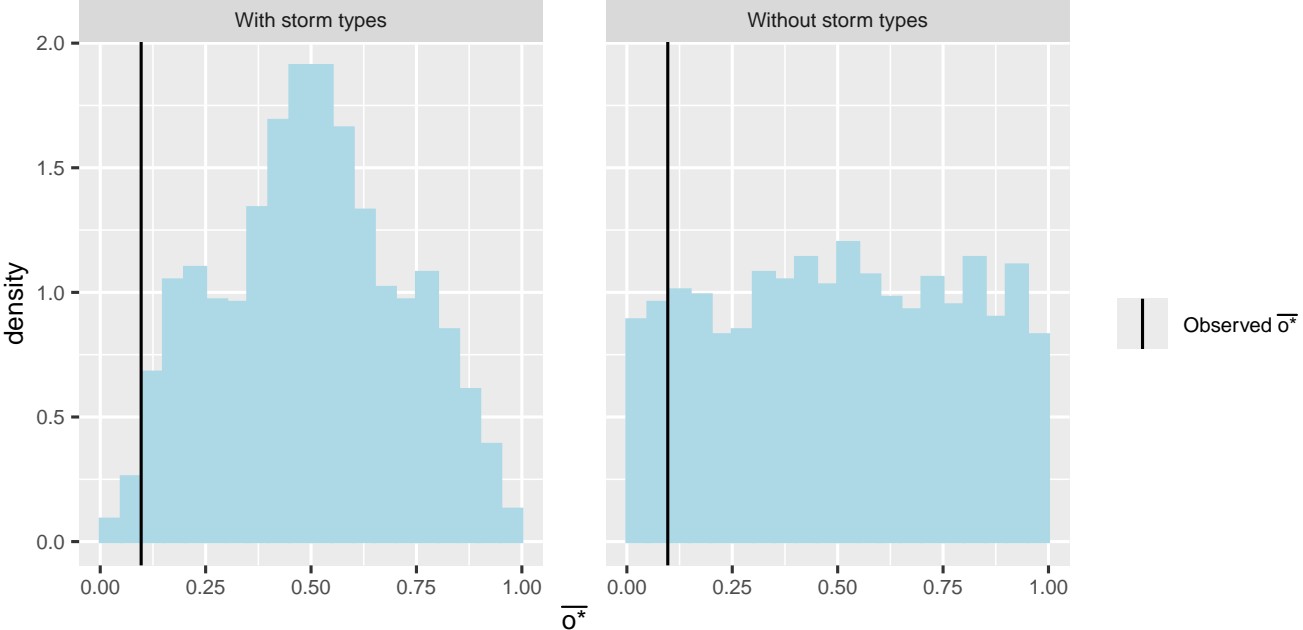

**Figure 4.** Prior predictive check for the occurrence model, the time average of prior predicted occurrence, $\bar{o}^*$, for the Atmospheric model with and without storm types.

only difference in prior choice between the regression parameters of covariates conditioned on storm types and the regression parameters of covariates that are not.

## 3.2 Posterior distribution

The posterior distributions for the atmospheric regression parameters can be seen in Figs. 6 and 7 for the occurrence and amounts models respectively. There is a relatively large amount of overlap between the posterior distributions of the parameters without storm types (w/o) and the parameters given a particular storm type and also between storm types. There are some differences between parameter values for the different storm types however, and within a storm type, the different stations have different regression parameter responses. One of the advantages of using named storm types is any potential physical interpretation of their parameter values. A full investigation into the relationship between the storm types, precipitation and the atmospheric variables is beyond the scope of this study, but an illustration of what could cause these physical relationships is valuable. In general, how precipitation is related to an atmospheric variable will depend on the storm type present.

Taking an example from the occurrence model (Fig. 6), compare the w500 regression parameters for storm types containing fronts but not thunderstorms (FO, CF) to the parameters for storm types containing thunderstorms but not fronts (TO, CT). When a front but not a thunderstorm is present the regression parameter for vertical velocity is more strongly negative than when a thunderstorm but not a front is present. When a front is present, the vertical velocity may be more important for




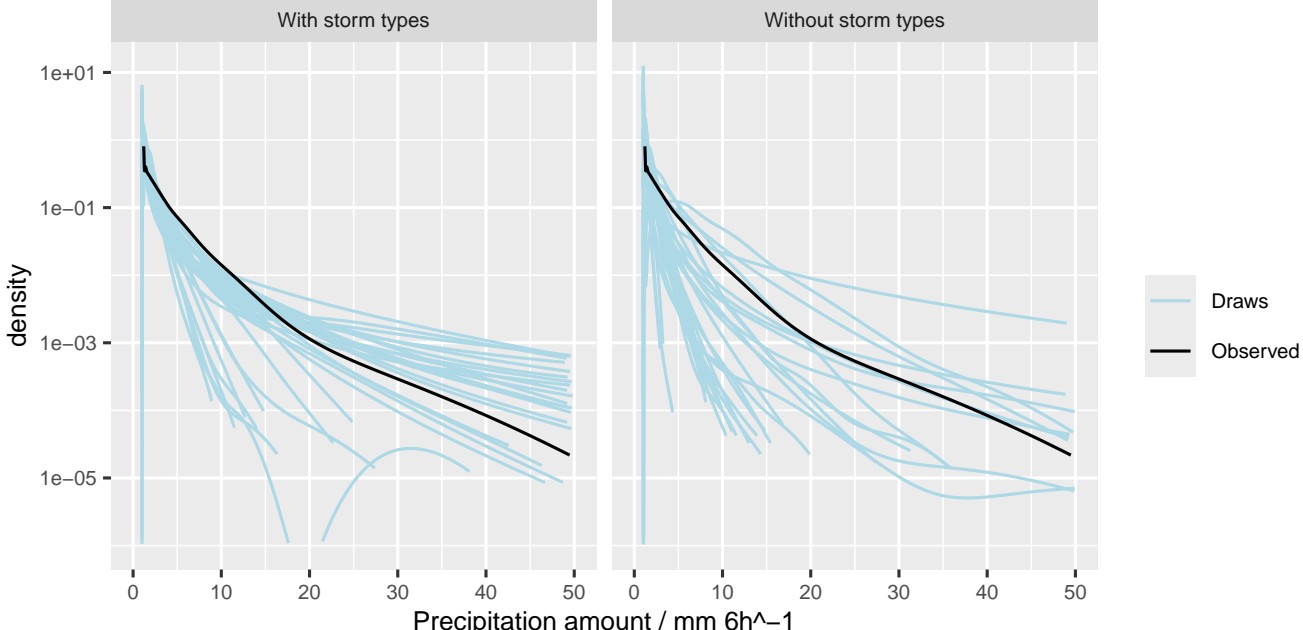

**Figure 5.** Prior predictive check for the amounts model, draws from the PrPD, for the Atmospheric amounts model for Wattisham. Kernel density estimates made using the logKDE package (Jones et al., 2018).

determining whether precipitation will occur or not due to the frontal area covering the region both ahead of and behind the front. Conversely, when a thunderstorm is present the regression parameter for vertical velocity is closer to zero. This weak dependence on vertical velocity could be associated with the scale of the systems and precipitation features in question. The presence of a thunderstorm storm type indicates environments conducive to thunderstorms over a large area (Catto and Dowdy, 2021). The vertical velocity over a grid box of ERA5 resolution (approximately 31km) is unlikely to capture the scale of an

individual convective system over the station in question.

   Taking an example from the amounts model (Fig. 7), for the Plymouth and Tiree stations, the intercepts for the FO and CF storm types are large and the q850 regression parameters for the FO and CF storm types small. This implies that the presence of a front is explaining some of the variance in precipitation that is otherwise (in the absence of storm types) explained by specific humidity for these stations. This is possibly because Plymouth and Tiree are on the west of the UK where fronts bring

moisture from the Atlantic. Notably, the parameter values of the CO storm type have a large spread which overlaps with the parameter values for the model without storm types for most stations and regression parameters for both the occurrence and amounts model. This could be because of the large areas covered by this storm type. It is important to recognise the limitation in interpreting regression parameter like this. These statements are not conclusions, rather they are examples to illustrate the potential physical interpretation of the regression parameters conditioned on storm types.





**Figure 6.** The posterior distribution of the Atmospheric occurrence model for the three stations for the intercept and atmospheric variables w500 and r850. w/o refers to the posterior distribution of the Atmospheric occurrence model without storm types.



**Figure 7.** The posterior distribution of the Atmospheric amounts model for the three stations for the intercept and atmospheric variables w500 and q850. w/o refers to the posterior distribution of the Atmospheric amounts model without storm types.



## 3.3 Calibration

The Probability Integral Transform (PIT) histograms for the three station and three covariate choices with and without storm types are calculated for the overall model (Fig. 8) and for just the amounts model (Fig. 9). The amounts model (Fig. 9) only includes the PIT values of the cases when precipitation occurs in the data. There is little evidence of miscalibration in the overall model PIT histogram for all stations and covariates. However, the amounts model is clearly miscalibrated with an
over-prediction bias that is present for all stations and covariates. This is not improved by the inclusion of storm types.

Fig. 10 shows the comparison between draws from the PPD and the observed precipitation distribution for the Atmospheric model with and without storm types. Figs. S1 and S2 in the supplement show equivalent (and similar) plots for the Null and Seasonal models respectively. Similar to the PIT histogram there is a consistent miscalibration in precipitation amounts. The WG distribution underestimates the frequency of low precipitation amounts and overestimates medium to high precipitation
amounts. This miscalibration isn't improved by the inclusion of storm types.

## 3.4 Model comparison using scoring rules

In order to determine if storm types improve the occurrence or amounts model the Brier Score (BS) is used to quantify the predictive skill of the occurrence model and the CRPS to quantify the predictive skill of the amounts model (Amounts CRPS) and the overall model (CRPS). As discussed in Sect. 2.9 the BS and CRPS are fitted using 10 fold cross validation meaning
the scores are all determined on out of sample data. These scores can be seen in Fig. 11 for the three different stations.

The most complicated model, Atmospheric with storm types, has the smallest (best) average CRPS, BS and Amounts CRPS. The Null, Seasonal and Atmospheric models all have a reduction in average CRPS, BS and Amounts CRPS by conditioning parameters on storm types. The Atmospheric model is improved the least by the inclusion of storm types, with a CRPS difference of 0.012 mm $6h^{-1}$ averaged across the three stations. Any difference between the Null and Seasonal models is
comparatively very small with the exception of the Wattisham amounts model which saw an improvement in the average Amounts CRPS. The average CRPS difference between the Atmospheric model without storm types and the Null model without storm types is 0.096 mm $6h^{-1}$. The average CRPS difference between the Null model with storm types and Null model without storm types is 0.032 mm $6h^{-1}$. This means the improvement to the simplest model by including storm types is 33% of the equivalent improvement by including atmospheric covariates.
To determine if the score differences are significant a Diebold-Mariano (DM) test is performed (Diebold and Mariano, 1995). The improvement to the CRPS due to storm types for all three models (Null, Seasonal and Atmospheric) have DM test statistics well above 2.58 for all models and these can be seen in Table 4. For reference a DM statistic of 2.58 would correspond to a two-tailed p-value of 0.01 that the equal predictive skill hypothesis is true. The smallest DM test statistic was for the average Amounts CRPS difference between the Atmospheric models with and without storm types for the Wattisham
station with a value of 1.83. Taking an example where there was negligible improvement, the DM test statistic for the average CRPS difference between the Null and Seasonal models is 0.56, which corresponds to the large two tailed p-value of 0.58.





**Figure 8.** PIT histogram for the overall WG models with amounts conditional on occurrence.





**Figure 9.** PIT histogram for the amounts WG models only. It is the PIT of the model given that rain has occurred.





**Figure 10.** Plot showing draws from the PPD and observed precipitation for the Atmospheric model with and without storm types for the three different stations, for precipitation greater than 1 mm $6h^{-1}$. Note that this figure shows the overall model, but only the amounts greater than 1 on a log10 scale. This means that the effect of any false positive occurrence values would be present. The WG distribution is consistently miscalibrated with respect to the observed distribution.





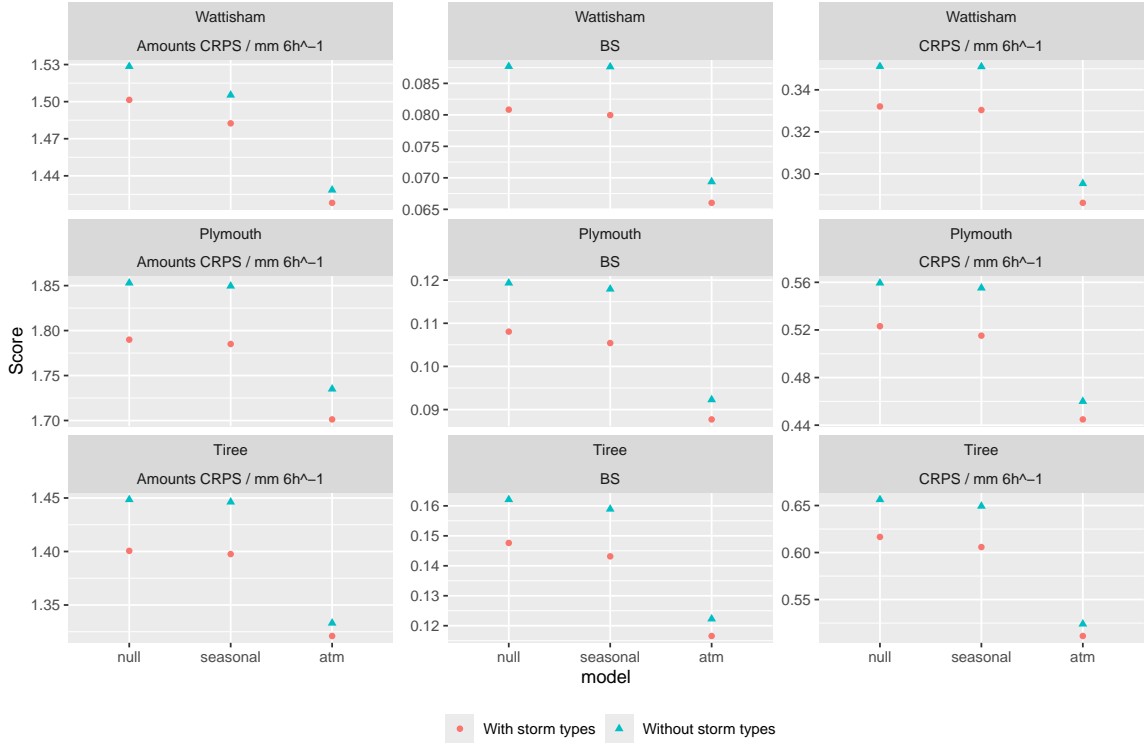

**Figure 11.** Average proper scores for the three weather stations and three covariate choices. The BS and amounts CRPS score the occurrence and amounts models respectively. The score of the overall model is labeled CRPS. The closer to zero the better the score.

**Table 4.** DM test statistic values for the three scores and models comparing the score with storm types to the score without storm types.

| Model | Score | Wattisham DM | Plymouth DM | Tiree DM |
|---|---|---|---|---|
| Null | BS | 14.06 | 18.54 | 15.63 |
| | CRPS | 12.64 | 16.53 | 14.84 |
| | Amounts CRPS | 4.19 | 9.76 | 9.86 |
| Seasonal | BS | 14.35 | 19.67 | 17.25 |
| | CRPS | 13.40 | 17.36 | 17.07 |
| | Amounts CRPS | 3.70 | 9.74 | 9.52 |
| Atmospheric | BS | 10.33 | 11.79 | 10.93 |
| | CRPS | 8.45 | 10.29 | 9.76 |
| | Amounts CRPS | 1.83 | 5.52 | 3.35 |





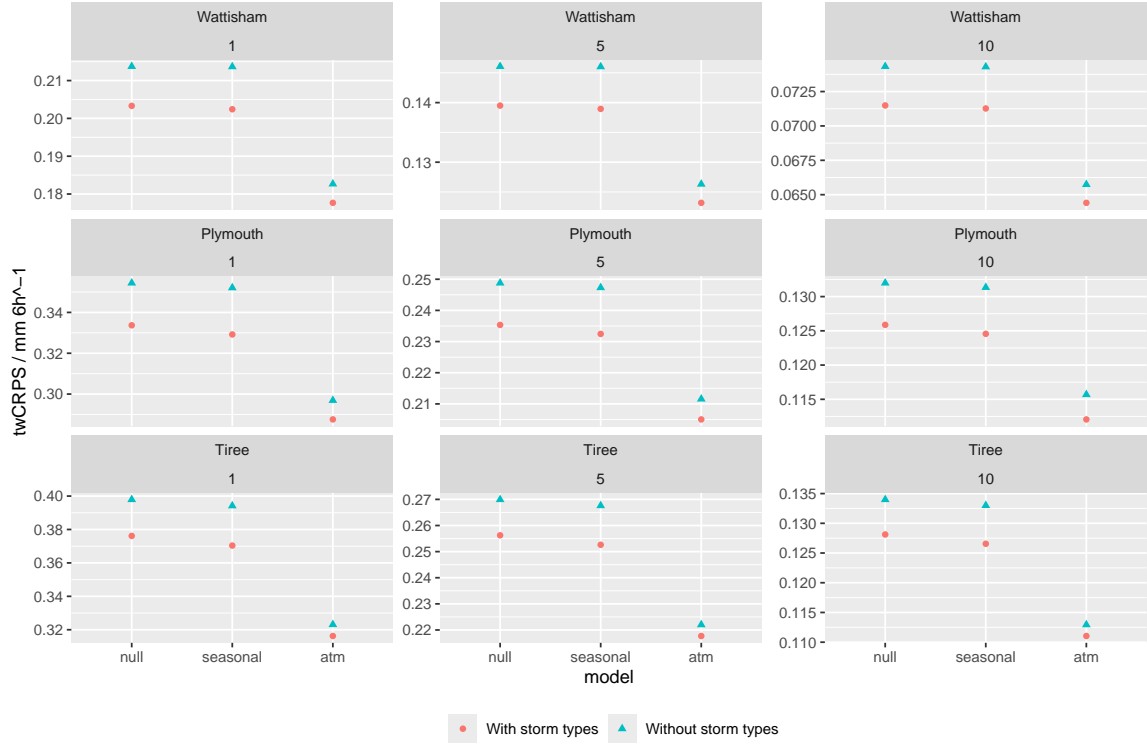

**Figure 12.** Average twCRPS values for the three stations and models. The three $\mu$ threshold values are: 1, 5 and 10 mm 6h$^{-1}$.

In order to assess the WG's ability to represent extremes the twCRPS values for the three stations, three covariates and for three different thresholds are calculated. These are plotted in Fig. 12. The threshold is the mean, $\mu$, of the normal CDF (the chosen weight function) in Equation 22. The standard deviation $\sigma$ of the weight function is chosen to be 7.5 mm 6h$^{-1}$. The three thresholds chosen are 1, 5 and 10 mm 6h$^{-1}$. The maximum precipitation amounts in the dataset for the three stations are 40.6, 38.6, and 43.0 mm 6h$^{-1}$ for Plymouth-Mountbatten, Wattisham and Tiree respectively.

The twCRPS leads to the same conclusion as the CRPS, with again the Atmospheric model being improved the least by an average twCRPS difference of 0.007, 0.005 and 0.002 mm 6h$^{-1}$ for the thresholds of 1, 5 and 10 mm 6h$^{-1}$ respectively. Higher thresholds were also investigated, but these lead to very small twCRPS values which are difficult to interpret, and to thresholds which only have a small number of observations above them.

## 4 Discussion and Conclusions

This study builds on previous WG work in four ways. First, a new precipitation GLM WG conditioned on discrete storm types with clear physical interpretation is proposed. Second, a Bayesian statistics framework is used with observationally informed priors for the WG. Third, model comparison is performed using the PIT histogram, draws from the PPD, and proper scoring





rules. The use of a proper threshold weighted score, the twCRPS, is explored for model comparison of precipitation WGs. Finally, the utility of using a DM test to quantify the statistical significance of average score differences is demonstrated.

A standard GLM WG with atmospheric covariates is improved, significantly at three stations, by conditioning the GLM regression parameters on storm types. The storm types improved the Atmospheric model by an average CRPS difference of 0.012 mm $6h^{-1}$, averaged across the three stations. This score difference is significant but small. Broadly this result is

consistent with the results of San-Martín et al. (2017) who similarly compared a GLM conditioned on weather types found by k-means clustering to one not conditioned on them. The most comparable metric in San-Martín et al. (2017) is the relative bias, the mean error between the observed and downscaled precipitation series as a percentage of the observed precipitation amount, which they found to be similar between the two WGs. Gutiérrez et al. (2019), a large statistical downscaling intercomparison project, also included a precipitation weather generator conditioned on k-means weather types in the intercomparison and

found a small improvement in mean wet-day precipitation bias (mean amounts given occurrence) when conditioning on the weather types. Of note is the improvement to the average CRPS from just conditioning on storm types without covariates (the Null model with storm types) which provided 33% of the equivalent improvement by including atmospheric covariates averaged over the three stations.

The use of a Bayesian framework for the WG has some consequences for the interpretation of the output precipitation and

provides an opportunity for modelling impacts. Draws from the PPD include both the parameter uncertainty and the internal variability from drawing from the likelihood function. This means different draws from the posterior distribution of the WG parameters produce different predicted time series of precipitation. Multiple precipitation time series can then be inputted into impact models to take into account parameter uncertainty in the WG model. We have tried to be transparent in the reasoning for the prior choices. In this case different priors have been chosen for the models with storm types and the models without

storm types. Instead of the same prior distributions, the priors are chosen to have a similar PrPD, a judgement that was made to avoid giving one model worse prior assumptions than another. Additionally, this means that priors are elicited based on an observable quantity, precipitation, rather than unobserved covariates. Assessments based on observed quantities benefit from an intuitive understanding of what is or is not physical and are considered best practice by some (Kadane and Wolfson, 1998). Other judgments are available, and indeed most WG intercomparisons do not face this problem as they are not using a Bayesian

model, but here we hope to be honest about this decision.

Here calibration is tested graphically using the PIT histogram and comparing the observed density to the PPD. Both calibration plots (Figs. 8 and 10) showed a consistent pattern of miscalibration. The error here is probably introduced by the choice of a Gamma distribution for the amounts GLM, and may be true for all Gamma precipitation WGs. Realistically, this is quite a simple choice and more flexible distribution choices might reduce this.

The use of the CRPS and BS to assess the skill of downscaling methods is relatively common (e.g. Rahmani et al., 2016; Sohrabi and Brissette, 2021; Krouma et al., 2022), but not standard practice. San-Martín et al. (2017) and Gutiérrez et al. (2019), for example, focus on mean and variance errors for the amounts of precipitation. Given the skewed distribution of precipitation, a WG could simulate the mean and variance of precipitation well and still fail to model the tail or shape of the distribution. Proper scores like the CRPS, therefore, are useful tools for WGs as they provide a rigorous statistic to compare observations



with the full distribution. DM significance tests then provide the confidence in any conclusions drawn from average proper score differences and should be included in any analysis using scoring rules.

The twCRPS is used here to explore the skill of the different WGs at larger values of precipitation. However, a key issue is apparent from the use of the twCRPS here. Score values below the threshold of the twCRPS are suppressed, and values above it are not. This means that the magnitude of a twCRPS score difference depends more on observations above the threshold than below. As the threshold increases there are progressively fewer observations above a given threshold than below it. For Wattisham, for example, the number of observations greater than or equal to $10 \text{ mm } 6h^{-1}$ is quite small at 165. A judgment based on weighting such a small sample size seems inappropriate, and similar concerns have been raised previously (Lerch et al., 2017), but a full investigation of the consequences of sample number above threshold in the twCRPS in this case is beyond the scope of this paper.

There are several avenues for further study. This WG is single site, but the storm types data is available for Europe, so a clear direction for future investigation would be to include storm types as an input to a multisite WG and to repeat this analysis for a larger domain. Additionally, this WG is a hurdle Gamma GLM, a relatively simple choice, so the effect of conditioning more complex WG models on storm types could be investigated. Here a focus has been using proper scores for model calibration and sharpness but other approaches could be taken, in particular evaluation of the output precipitation time series, for example by comparing consecutive dry day length. Finally, the storm types dataset is based on ERA5, but with an equivalent set of storm types calculated for CMIP6 models the WG could be used to downscale climate model output.

*Code and data availability.* A Rmarkdown file that shows an example of how to repeat the analysis used in this study, along with the data used, can be found here: https://doi.org/10.5281/zenodo.16795621 (Bell, 2025).

*Author contributions.* PB contributed to the methodology and interpretation of results, performed the analysis and verification, and wrote the original manuscript with contributions from all co-authors. JC provided the storm types dataset and contributed to the methodology and interpretation of results. SS contributed to the methodology and interpretation of results. SS, JC and AJ supervised the project.

*Competing interests.* The authors declare no competing interests.

*Acknowledgements.* With thanks ECMWF for the ERA5 reanalysis and the UK Met Office for the MIDAS Open station data. This research is supported by IBM United Kingdom Ltd. via EPSRC ICASE scheme grant code EP/W522156/1.



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
