# Peer review of "Developing and evaluating a Bayesian weather generator for UK precipitation conditioned on discrete storm types"

_EGUsphere, 2025_

## Referee Comment (RC1)

**Manuscript EGUsphere-2025-4110 'Developing and evaluating a Bayesian weather generator for UK precipitation conditioned on discrete storm types' by Bell et al. Referee's report**

This paper presents a single-site precipitation generator based on generalised linear models (GLMs). The claimed innovations are the use of a Bayesian framework for model fitting and simulation; the incorporation of weather types 'based on physical a priori information about storms' (line 47); and the use of scoring rules for performance assessment.

In principle, some of these innovations are interesting. Unfortunately however, I don't think the paper makes the case for them strongly enough at present. Moreover, it reads more like a standalone data science exercise than a HESS paper because (i) it doesn't engage adequately with the hydrological requirements of a weather generator; (ii) there's little attempt to interpret the results except for some rather speculative comments on lines 336–354; and (iii) there's a lot of mathematics which is sometimes superfluous and which obscures the narrative (especially in Sections 2.3 to 2.8, which contains a lot of textbook material without any real rationale).

A further critical issue is that the paper does not explain why another precipitation generator is needed. There is an enormous literature on this topic, and the best generators currently available have impressive capabilities — albeit with some known deficiencies. To justify introducing yet another one, there is a need to identify a gap — for example in terms of capability, performance, data requirements, ease of implementation, or computational speed; and to demonstrate that the proposed approach goes some way towards filling that gap. No such gap is identified here, with the possible exception of the approach to weather typing (but see comments on this below); and there is no attempt to compare the proposed method with any credible competitors, so it's impossible to evaluate whether practitioners should start using it. Indeed, the performance hasn't been evaluated in much detail at all — for example, there's no attempt to explore the ability of the generator to reproduce the kinds of properties that are important in hydrological applications. From the information that is provided however (e.g. Figures 9 and 10), the performance seems disappointing to me: it's not hard to find approaches in the literature that reproduce the wet-period distribution much better than this. Some caveats to this assessment, however, are (i) that PIT histograms are not necessarily an appropriate way to assess the performance of a weather generator because the aim is not prediction (more on this below) (ii) the temporal resolution of almost all published weather generators is daily rather than six-hourly as in the present paper, and it's possible that daily results from the literature are not a good guide to what's possible at six-hourly resolution.

The reservations above are fundamental in my view. I will summarise a few other 'major' comments, followed by more detailed points that may be useful in any revision of the manuscript.

**Major points**

Bayesian approach: what's the rationale for using a Bayesian approach? As acknowledged on lines 424–425, it's fairly unusual in the literature on weather generators and, although it has some conceptual advantages (e.g. the ability to incorporate parameter uncertainty into the simulated sequences as described in lines 415–418), it also has some potential disadvantages relating to prior choice, difficulty of implementation and computational time (the latter issue should be mentioned, incidentally — how long does it take to fit and simulate these models?). Given the state of the field, this choice needs clear justification I think.

Actually, as someone with considerable experience of GLMs in the context of weather generation, I have often wondered whether a Bayesian approach would be worthwhile as a way of propagating parameter uncertainty into the simulated sequences. My gut feeling is that in most cases it probably wouldn't make much difference, because the datasets available for model calibration are usually large so that parameter uncertainty is small. I don't *know* this, however, and I think a comparison would be an interesting and useful exercise: fit the same models in a 'classical' way (i.e. use the maximum likelihood estimates of the parameters to generate the simulated sequences) and a Bayesian way in which you sample from the posterior predictive distributions, and compare the two in terms of both performance and computational cost.

Some other comments on the Bayesian material are as follows:

- 1. The approach to choosing priors is potentially interesting, but (as elsewhere in the paper) there isn't enough detail to figure out exactly what has been done. E.g. on lines 189–190 there is a statement that "each regression parameter is given the same prior regardless of storm type", which I interpreted as meaning that the same prior is used for all coefficients and storm types. Table 3 shows, however, that this is not the intended meaning: rather, each coefficient has its own prior standard deviation, and these standard deviations are common to all storm types. In this case, though, there must be multiple combinations of prior standard deviations that would give similar histograms / densities to those shown in Figures 4 and 5: how exactly were the final choices arrived at, therefore?
  - I also don't understand how Figure 5 has been produced. Why are there multiple blue curves, and why does one of them seem to have a break in it at around 20mm / h at the bottom of the left-hand panel? Why do the curves have different horizontal extents? Surely a given prior choice should produce a single prior predictive distribution?
- 2. Whenever informative priors are used, as here, there is a need either to provide a plausible justification for the choices that have been made or to examine sensitivity to alternative 'reasonable' specifications. Neither has been done here: the criteria set out in lines 310–314 are rather vague, and no attempt has been made to investigate the sensitivity to these choices. Also, Figure 4 seems to contradict the assertion, on line 311, that the prior choices yield a range of  $\overline{o}$  between 0.1 and 0.3.
  - In any case, I would be surprised if informative priors are needed: GLM-based weather generators are usually fitted using maximum likelihood without any problem, which suggests that the data should be sufficiently informative to identify the parameters based on the likelihood alone: flat or non-informative priors should work fine therefore, which would eliminate the need to worry about the issues just mentioned.
- 3. The presentation of the Bayesian methodology contains both too much and too little detail. There's too much in lines 143–164, where none of expressions (7)–(11) is needed to understand the subsequent development: the reader just needs to know that the required quantities are obtained as samples from the relevant posterior distributions using the brms package. On the other hand, nothing is said about convergence diagnostics, numbers of chains, iterations and burnin etc., which are essential to provide reassurance that the Markov Chain Monte Carlo techniques have been used appropriately.

**Performance assessment:** the scoring rules used to assess performance are designed for use in forecasting settings, where there is a one-to-one correspondence between each forecast distribution and the corresponding observation. But weather generators are not forecasting systems: rather, the aim is

to produce simulated sequences with the correct statistical properties over some time period. Of course, day-by-day forecasting is one way to achieve this; but it's not the only way to achieve it and, for applications in which weather generators are used (e.g. impacts studies), forecast performance is not relevant. Also, the scoring rules considered here only allow an assessment of *relative* performance (i.e. to say that models using storm type information are better than those which don't). By contrast, in hydrological applications (given that the paper is submitted to HESS), one typically wants to know that hydrologically relevant features of the precipitation regime are reproduced acceptably in absolute terms. Maraun et al. (*Earth's Future*, 2015, doi: 10.1002/2014EF000259) provide guidance on performance assessment for downscaling methods in general, including weather generators.

**Approach to weather typing:** lines 46–47 of the paper suggest that the use of physically-motivated weather types in weather generators is novel. I would be surprised if this is true. I can believe that the Catto and Dowdy typology hasn't been used before; but it's hard to imagine that someone hasn't previously constructed a precipitation generator using, for example, the Jenkinson-Collison weather types which are also 'physically-informed' (see DOI 10.1007/s00382-022-06658-7 for a recent reference).

On a related point, lines 36–40 mention some difficulties with the use of empirically-determined weather types in conjunction with climate projections, arising essentially because climate model biases could lead to a different clustering structure compared with that found in observations. At the end of the paper, there is a suggestion that the proposed method could be used in conjunction with CMIP6 output — but no acknowledgement of the fact that this could be subject to its own difficulties because the types are based on 'thresholds in atmospheric variables' (line 49). Any index based on thresholds is notoriously difficult to handle in the presence of climate model biases; and, presumably, the categorisation used here also reflects inter-relationships between atmospheric variables which makes it even trickier. Section 11.5 of Maraun and Widmann (2018, in the authors' reference list) discusses the relevant considerations when choosing atmospheric information for use in downscaling studies; see also Wilby et al. (2004).

**Model structure and diagnostics:** the structure of the models considered (e.g. Table 2) is very simplistic. The most obvious omission is any consideration of temporal dependence. It's possible that this is captured by the storm types; but, in my view, this is unlikely and no attempt has been made to check it. For hydrological applications, it's essential that weather generators can reproduce autocorrelation and persistence in precipitation.

The representation of seasonality via a single sinusoid is also probably oversimplistic: there should at least be some checks for residual seasonality which may, for example, require the inclusion of further harmonics. These points — and, indeed, a range of diagnostics that can be used to check the formulation of a GLM-based weather generator — are all discussed in several of the papers that the authors cite.

**Critical evaluation:** the paper is short on critical evaluation, especially in the discussion section which is rather generic and offers few real insights.

**Minor / detailed points**

- The title claims that it's a weather generator, but it would be more correct to describe it as a precipitation generator: weather generators usually produce multivariate output.
- The paper is acronym-heavy in places, which hinders readability.
- Lines 31–32: this review of weather-typing approaches is limited and omits an important class of approaches based on nonhomogeneous hidden Markov models, which are arguably more defensible because the identified weather types are directly linked to the associated precipitation.
- Lines 58–9: it is quite confusing to say that the prior predictive distributions are 'based on the observed predicted distribution for precipitation', because this suggests that you're using the observed precipitation distribution to choose the priors. The choice is based on the *implied* distribution of precipitation for each candidate prior.
- Lines 66–69: as noted above, the study compares models with and without storm types but there's no comparison to any other credible approach, which is needed to demonstrate that the paper makes a worthwhile contribution to the field.
- Lines 110–111: given the substantial amount of literature that uses separate occurrence / intensity models for precipitation, I don't think it's necessary or helpful to mention hurdle models. The precipitation literature goes back way further than they do: if anything, public health research should be referencing the precipitation literature, not the other way round!
- Line 140: by fitting a separate set of regression parameters for each storm type, the model effectively includes interaction terms between storm type and each of the other covariates in the model. This is parameter-intensive, probably not justified, and seems unnecessary given that interactions including with categorical covariates can be accommodated easily within the standard GLM setup. I see no good reason for treating the different storm types separately, therefore: the available information could probably be used more effectively by fitting a single model with interactions that would enable investigation of (for example) whether some covariate effects are common to all storm types whereas others vary.

This comment also relates to the discussion of Figures 6 and 7, on lines 330–332.

- Line 165: why only these three model structures? GLM-based statistical modelling usually involves a much more nuanced approach to model building and selection. See major points above.
- Lines 177–184: the description of Figures 2a and 2b does not match what I see in those figures (even after I realised that paler colours represent higher values, which seems very unintuitive and took me some time to notice). At the very least, a loess smooth should be superimposed on the plots so that any systematic structure can be seen.
  - Moreover, the discussion of Figures 2c and 2d suggests that the distinction between wet and dry periods is much clearer than is actually the case. E.g. it is by no means exclusively true that 'occurrence happens when w500 is more negative' as claimed. This all needs to be tightened up, to reflect what the data are saying.

- Lines 194–201: the role of this discussion of prior predictive distributions is unclear. It would perhaps make more sense if the material currently in Section 3.1 could be moved in here: after all, that material is not really about 'results' but rather about prior choice. Either way though, the explanation of how the priors were chosen needs to be clearer as noted in my major comments above and I don't think equations (14) and (15) really help. If my understanding is correct, all you're doing is taking a bunch of samples from the priors and then, for each one, taking large samples from the corresponding occurrence and amounts models to get a corresponding empirical precipitation distribution: no equations are needed to say that.
- Lines 203–204 "a common tool to evaluate forecasts against observations": as per major comments above, weather generators are not forecasting systems.
- Lines 240–241: it may be worth noting that the Diebold-Mariano test has become the standard test for equality of forecast performance when using scoring rules. As presented, it comes across as an innovation in the paper.
- Line 243: how realistic is it to assume that the sequence of score differences is covariance-stationary? Why is the potential for autocorrelation recognised here, but not anywhere else in the paper?
- Lines 280–281: this description of Hamiltonian Monte Carlo won't help a reader who hasn't encountered it before, and isn't necessary for a reader who has encountered it. As noted previously, all that's necessary is to indicate what software was used and to provide details of convergence diagnostics etc.
- Lines 290–291: this random subsetting is another place where the potential for temporal dependence could be problematic. In general, cross-validation for time series needs careful thought.
- Lines 316: what "Gamma prior" is being referred to here? If it's for the shape parameter of the amounts model, then the prior mean is 2 which is probably much too high. My experience is that a shape parameter of 0.6–0.7 is usually appropriate for daily rainfall in mid-latitudes, and I would expect it to be even lower at a subdaily timescale.
- Lines 320–322: here is an acknowledgement that the informative priors are not consistent between models. This is related to the concerns in my major comments above, about the use of informative priors in general.
- Figures 6 and 7: captions should be "The posterior distributions of the regression coefficients in [the respective models]".
- Lines 359–360: here is an acknowledgement that the amounts model is a poor fit, but there's no attempt to investigate it. The problem is either due to the assumed gamma distribution for precipitation amounts, or to a poor choice of covariates: I suspect primarily the latter, given the track record of gamma-based GLMs in this kind of application elsewhere. In any case though, Figure 9 isn't an appropriate way to check the performance of a weather generator because it treats the outputs as forecasts (see major comments again).
- Lines 416–417 "different draws ... produce different ... time series": this is true of any other weather generator as well. What's important here is to determine how much of a difference it makes to incorporate parameter uncertainty into the sampling, and this hasn't been investigated.

- Lines 424–425: there's no such thing as a Bayesian model. The model is a model: it's the inference that's Bayesian (and, as noted earlier, the approach should be justified).
- Line 431: sorry to go on about this, but the reason that scoring rules aren't standard practice in this kind of application is because weather generators are not designed for forecasting.
- Lines 441–444: I don't understand the point being made here about small numbers of observations. The significance test accounts for this automatically, via the divisor in equation (25). It's possible that the concern relates to the power of the test which will probably decrease as the sample size decreases or, maybe, to the adjustment for potential autocorrelation which will become inaccurate due to poorly estimated autocorrelations (but which will also be less of an issue as the threshold increases, because the remaining observations will tend to become less dependent). It's not articulated clearly, however.
- Line 446: I think a multi-site extension of this methodology would be extremely difficult, due to
  the need to consider inter-site dependence in a Bayesian framework which requires full specification of the joint multi-site distribution. My own 2020 paper (doi 10.1016/j.envsoft.2020.104867)
  discusses some of the challenges and available options. If the authors think it's feasible using their
  methodology, they need to give an indication of how it might work.

Richard Chandler University College London 7th October 2025

---

## Referee Comment (RC3)

Review. Developing and evaluating a Bayesian weather generator for UK precipitation conditioned on discrete storm types by Paul Bell et al.,

**Overview.**

This paper presents a Bayesian Generalized Linear Model (GLM) weather generator for UK precipitation, conditioned on a newly developed dataset of physically based storm types (cyclones, fronts, and thunderstorms). The authors evaluate the model using proper scoring rules (CRPS, BS, twCRPS) and introduce the Diebold–Mariano (DM) test to assess the statistical significance of performance. Results show that conditioning on storm types significantly improves predictive skill, though the magnitude of improvement is modest. The Bayesian framework allows for transparent uncertainty quantification, and the approach demonstrates good potential for extending WG applications to process-based climate downscaling. However, the paper also highlights calibration issues, with consistent overprediction of moderate-to-heavy rainfall, suggesting room for refinement in the model's distributional assumptions.

→ Overall, the paper is a well-structured and innovative study that advances both the methodological and interpretive aspects of weather generator development, especially in linking atmospheric processes with statistical modeling. However, some minor clarification are needed.

**Questions.**

- 1. The PIT histograms and posterior predictive checks reveal a consistent overestimation of moderate-to-high precipitation amounts. The discussion could identify whether this bias stems from the Gamma assumption, the linear predictor structure, or the aggregation timescale (6-hourly).
- 2. The inclusion of storm types improves CRPS by  $\sim 0.012$  mm  $6h^{-1}$ , a statistically significant so far small gain. The authors should clarify whether this difference has practical consequences for hydrological or risk modeling applications, or whether it primarily demonstrates methodological robustness.
- 3. The threshold-weighted CRPS (twCRPS) is an interesting choice for assessing extremes, but its sensitivity to limited high-end samples is a concern. The authors could consider (or at least discuss) complementary extreme metrics, such as quantile skill scores or percentile biases, to confirm the robustness of conclusions for heavy precipitation.
- 4. One of the motivations of this paper is to apply the WG to GCM projections, it would be useful to discuss how storm-type classification might perform on biased model fields, and whether threshold recalibration or bias correction would be required.

**Minor comments.**

- L.13: Continuous Ranked Probably Score (CRPS) → Continuous Ranked Probability Score (CRPS).
- L.51: meterological → meteorological.
- L.172: where we expect → where we expect.

Figure 4 caption could clarify what "Observed o\*" represents (is it mean observed occurrence?). In general, some figure captions are long and could be more concise.